# Influence of Polymer Film Thickness on Drug Release from Fluidized Bed Coated Pellets and Intended Process and Product Control

**DOI:** 10.3390/pharmaceutics16101307

**Published:** 2024-10-08

**Authors:** Marcel Langner, Florian Priese, Bertram Wolf

**Affiliations:** 1IDT Biologika, Am Pharmapark, 06861 Dessau-Roßlau, Germany; marcellangner@gmx.net; 2Department of Applied Biosciences and Process Engineering, Anhalt University of Applied Sciences, Bernburger Straße 55, 06366 Köthen, Germany; bertram.wolf@hs-anhalt.de

**Keywords:** Wurster fluidized bed coating, coated pellets, controlled drug release, spatial filter velocimetry

## Abstract

Background/Objectives: Coated drug pellets enjoy widespread use in hard gelatine capsules. In heterogeneous pellets, the drug substance is layered onto core pellets. Coatings are often applied to generate a retarded release or an enteric coating. Methods: In the present study, the thickness of a polymer coating layer on drug pellets was correlated to the drug release kinetics. Results: The question should be answered whether it is possible to stop the coating process when a layer thickness referring to an intended drug release is achieved. Inert pellets were first coated with sodium benzoate and second with different amounts of water insoluble polyacrylate in a fluidized bed apparatus equipped with a Wurster inlet. The whole process was controlled in-line and at-line with process analytical technology by the measurement of the particle size and the layer thickness. The in-vitro sodium benzoate release was investigated, and the data were linearized by different standard models and compared with the polyacrylate layer thickness. With increasing polyacrylate layer thickness the release rate diminishes. The superposition of several processes influencing the release results in release profiles corresponding approximately to first order kinetics. The coating layer thickness corresponds to a determined drug release profile. Conclusions: The manufacturing of coated drug pellets with intended drug release is possible by coating process control and layer thickness measurement. Preliminary investigations are necessary for different formulations.

## 1. Introduction

The drug dissolution of solid formulations depends on the solubility and dissolution rate of the drug substances and—in case of release—a number of parameters additionally influence the release kinetics: for example, drug substance interaction with formulation components (excipients), compression force and hardness in case of tablets and kind of binder in granulates, pellets and generally in polymer coatings. The knowledge of the dissolution rate and the release kinetics is essential and indispensable for optimum pharmacotherapy. Dissolution of solid substances runs approximately with first order kinetics due to diffusion processes. Certain drug formulations show zero order release with equal amounts released in equal time intervals. The superposition of several processes, for example, wetting of the solid drug dosage form, dissolution of the solid drug substance, diffusion of the drug molecules out of the dosage form, swelling of the dosage form in case of matrix formulations and swelling and water uptake of insoluble films leads to kinetic processes not meeting unambiguous zero or first order or square or cubic root equations. The release data are linearized by several models to evaluate the best approach. The coefficient of determination CoD of the linearized curve gives hints to the best adaptation and to the probability of the dominating process [1,2,3,4,5,6].

The model-independent parameters difference factor f_1_ and similarity factor f_2_ are used for release profile comparison; f_1_ describing the relative error between two release profiles calculated from the cumulatively released amounts at a certain time T for a test and a reference formulation or in general between two formulations—for example in the drug’s development. f_2_ is based on the sum of deviation squares of the released drug amounts of two release profiles [4,5,7,8,9].

Increasing attention is given to drug-loaded pellets and their release control by slowly swelling matrix systems or a final functional coating. The release of drug substances from matrix pellets prepared by extrusion/spheronization and finally coated with different amounts and types of insoluble ethylcellulose has been investigated [10,11,12]. Other authors report on the influence of the filler type on the drug release [13], the effect of the pH value of the release fluid [4], the storage conditions of drug and methylcellulose matrix pellets [14], the amount of enteric polymer coating [15] and the salt concentration of the release fluid [16]. The influence of talc and hydrogenated castor oil on the dissolution behaviour of metformin-loaded matrix pellets with an acrylic-based sustained release coating [17], the sustained release of Lisinopril from mucoadhesive matrix pellets [18] and the sustained-release of sinomenine hydrochloride from pellets manufactured by a novel whirlwind fluidized bed process have all been investigated [19].

Drug-layered inert pellets coated with polymer (heterogeneous pellets) were investigated in a similar way regarding the influence of the release kinetics by different modifications of the ethylcellulose coating [20], by ethylcellulose mixed with different amounts of polyvinylpyrrolidone (PVP) as pore former for controlled drug release [21], by alternating layers of ethylcellulose and polyvinylacetate [22], by various ethylcellulose coating levels and final curing [23] and by ethyl cellulose coating of acetaminophen-layered sugar pellets [24]. With a polyacrylate coating, the drug release from layered pellets was found to be retarded [7,25]. Variation of polymer type and layer thickness permits the control of the release rate in a wide range [8].

In our own earlier investigations heterogeneous pellets were manufactured by fluidized bed technology with a Wurster inlet. Inert microcrystalline cellulose pellets were coated with excipients and the easily water-soluble model drug sodium benzoate [26,27,28]. These sodium benzoate pellets (SB pellets) exhibited a narrow particle size distribution, high sphericity and homogeneous layers and gave very quick sodium benzoate release. For retarded release, the SB pellets were coated in a second step with different amounts of ethylcellulose, once more by fluidized bed technology with a Wurster inlet [29]. The release rate diminished with increasing layer thickness, as expected. Furthermore, the fluidized bed processes were controlled by in-line particle size measurement with the spatial filter velocimetry SFV probe [27,28] for process control regarding particle size, particle size distribution and ethylcellulose layer thickness.

The aim of the present project was the manufacturing of heterogeneous pellets in the fluidized bed with a Wurster inlet, the control of the process by in-line particle size and coating layer thickness measurements, the investigation of the kinetics of sodium benzoate release considering different kinetic models, the interpretation of the partial processes involved in the release, the correlation of the release rate with the polymer layer thickness and the detection of the coating process endpoint for the improvement of the pellet product quality regarding controlled drug release. For drug pellet manufacturing, a similar experimental approach as the one in [26,27,28] was chosen. Relatively small initial inert pellets (Cellets^®^175, median 170 µm), referring to a large specific surface area, were coated with a solution of sodium benzoate and a low amount of the water soluble PVP as a binder to improve the mechanical stability of the layer. In a second step (see [29]), the SB pellets were coated with different amounts of insoluble but slowly swelling polyacrylate for retarded release. The risk of undesirable agglomeration of those small pellets during sodium benzoate and polymer coating was practically eliminated by adjusting the process parameters and adding talcum to the coating fluid as an antistick agent. The SFV probe was installed for in-line particle size measurement and detection of agglomerates. The drug release was investigated and discussed applying zero order, first order, square root and cubic root equation kinetic models. Finally, the most probable release kinetics model was identified by the calculation of curve parameters—area under the curve (AUC), dissolution efficiency (DE) and mean dissolution time (MDT)—and by comparison of the CoD of the different kinetic models. The difference of the release profiles and the similarity of different polyacrylate layered pellet lots were calculated by the difference factor f_1_ and similarity factor f_2_. The linearization approach of the dissolution profiles is suitable for first order kinetics; for other release profiles a nonlinear approach can describe the dissolution curves more accurately and with a smaller standard deviation of the fitting parameter than the linearization-based calculation method [30].

## 2. Materials and Methods

### 2.1. Materials

Pellets of microcrystalline cellulose (Cellets^®^175, particle size range 150–200 µm, median 170 µm, IPC, Dresden, Germany), sodium benzoate (Applichem, Darmstadt, Germany, solubility 57 g in 100 g water at room temperature), PVP (Kollidon^®^25, Carl Roth, Karlsruhe, Germany), talcum (Talkum Pharma, C. H. Erbslöh, Krefeld, Germany), magnesium stearate (VEG Pharma, Rome, Italy) and polyacrylate dispersion (Eudragit^®^NE 30D, Evonik Industries, Darmstadt, Germany, containing ethylacrylate- methylmethacrylate copolymer 30% *w*/*w*) were used. All substances conform to European Pharmacopoeia (Ph.Eur.) quality [31].

### 2.2. Formulation of Sodium Benzoate-Coated Pellets

Microcrystalline cellulose pellets were coated with an aqueous solution of sodium benzoate 30% (*w*/*w*), PVP 1.5% (*w*/*w*) and talcum 0.5% (*w*/*w*) in a first coating step (Table 1). Sodium benzoate and PVP were dissolved one after another in purified water and finally talcum was suspended under agitation with a blade stirrer.

### 2.3. Formulation of Polyacrylate-Coated Sodium Benzoate Pellets

In a second coating step, SB pellets were layered with polyacrylate in three different concentration lots P1 (11.1% *w*/*w* PVP), P2 (14.3% *w*/*w*) and P3 (17.6% *w*/*w*). Magnesium stearate and talcum were added to the coating fluid as a plasticizer and an antistick agent, respectively (Table 2). The polyacrylate coating fluid contains a polyacrylate copolymer 13.3% (*w*/*w*), magnesium stearate 1.3% (*w*/*w*) and talcum 5.3% (*w*/*w*). A Eudragit^®^NE 30D dispersion was added to a beaker, and magnesium stearate and talcum were added one after another under strong agitation and homogenization by a disperser (Ultra Turrax T50 standard, Janke & Kunkel, IKA Labortechnik, Staufen, Germany, disperser length 225 mm, diameter 18 mm, rotation 5000 rpm).

### 2.4. Fluidized Bed Pellet Coating

The coating process was performed in a batch laboratory fluidized bed apparatus (GPCG 1.1, Glatt, Binzen, Germany) with a Wurster inlet and SFV probe installed into the process chamber [27]. A 1.0 mm diameter spray nozzle with a nozzle cap position of 2.5 scales was used. The distance of the lower end of the cylinder from the perforated bottom plate B was fixed to 20 mm. The process air volume rate was variable and adapted to the increasing weight of the pellets during the coating process in a range of 40–60 m^3^/h.

Cellets^®^175 were coated with the sodium benzoate/PVP/talcum aqueous fluid in a first step (Table 3). The second coating step with polyacrylate dispersion was performed under smooth conditions (lower spray rate and reduced process air temperature) to avoid the risk of the pellets adhering and sticking together. The polyacrylate coated pellets were finally tempered (one hour, 30 °C) in a tray dryer as a thin layer on a steel dish for coalescence and film forming completion.

### 2.5. Particle Size Coating Layer Thickness Measurement with SFV Probe

The particle size and thickness of the coating layer were measured in-line by the SFV probe (IPP 70, Parsum, Chemnitz, Germany). The probe was directly installed into the down-bed zone of the process chamber. Details of probe measurement are described elsewhere [27,28].

### 2.6. Sodium Benzoate Release and Content Investigation

The release was investigated with the dissolution tester (PTW 2, Pharmatest, Hainburg, 6 vessels, 1.0 L purified water, 37 °C, blade rotation 75 rpm). The sampling was performed after 10, 20, 30, 45, 60, 120 and 180 min. Samples were withdrawn and refilled by purified water. Sodium benzoate was analysed with a UV–Vis-Spectrophotometer (Spekol 1300, Analytik Jena, Germany, 1 cm quartz cuvette, wave length 220 nm).

For the sodium benzoate content investigation, 50 mg pellets (13.5 mg of which being sodium benzoate) were dispersed in 1.0 L purified water. The sodium benzoate dissolution and release were proved to be complete after 4 h and the content was analysed as above.

### 2.7. Linearization of Release Curves

The evaluation of the release curves was performed according to the different models of release kinetics also used by a number of authors [1,3,5,6,9,14]. In the first step of the release evaluation, the amount of cumulative released substance is plotted versus time. Linear curves arise in the case of zero order kinetics, i.e., equal amounts of the drug are released in equal time intervals (Equation (1)).
M_t_ = −k_0_ ∗ t + M_0_(1)

First order release kinetics are typical for the release of slightly soluble drugs from solid preparations like tablets, pellets and granules dominated by slow dissolution and diffusion control. The release rate is highest at the beginning of the process, according to the large concentration gradient being the most important factor in Fick’s first law for the transport flow density by diffusion (Equation (2)), and diminishes with a decreasing concentration gradient in the course of the process.
1/A ∗ dn/dt = −D ∗ dc/dx(2)

The released amount M_t_ at the moment t refers to (Equation (3)), and linearization results in the Sigma minus function (Equation (4)).
M_t_ = M_0_ ∗ (1 − exp(−k_1_∗t))(3)
ln (M_0_ − M_t_) = ln (M_0_) − k_1_ ∗ t(4)

The Weibull function (Equation (5)) and its linearized form (Equation (6)) presuppose first order kinetics.
M_t_ = M_0_ ∗ (1− exp (−t ^b/a^)(5)
ln (−ln (1−M_t_/M_0_)) = b ∗ ln (t) − ln (a)(6)

Square root kinetics occurs at non-disintegrating solid matrix formulations (Equation (7)).
M_t_ = k_q_ ∗ √ t(7)

Cubic root kinetics are observed in the case of spherical multiparticulate formulations (linearized form, Equation (8)).
M_t_^⅓^ = M_0_^⅓^ − k_c_ ∗ t(8)

### 2.8. Model Independent Parameters: Difference Factor f_1_ and Similarity Factor f_2_

The difference factor f_1_ describes the relative error between two release profiles calculated from the cumulative released amounts R_i_ and T_i_ at distinct moments for the reference and test formulations (Equation (9)). The similarity factor f_2_ is based on the sum of deviation squares of the released drug amounts (Equation (10)) and describes the statistical similarity between two release profiles. The value is 100 in case of identical profiles and 50–100 for similar profiles. Both factors are used to compare the release profiles of generic and standard drug product in order to decide whether the profile of the generic drug product surpasses that of the standard. In this study, both factors are used to evaluate the differences and similarities between sodium benzoate release profiles with different polymer coatings.
(9)$$f_{1}=\frac{\sum_{i=1}^{n}\left|R_{i}-T_{i}\right|}{\sum_{i=1}^{n}R_{i}}*100$$
(10)$$\;f_{2\;}=50*log\;\left\{{\left[1+\frac{1}{n}*\overset{n}{\underset{i=1}{\sum }}{\left(R_{i}-T_{i}\right)}^{2}\right]}^{-0.5}*100\right\}$$

### 2.9. Microscopically Investigation

Coated pellets were placed on black paper for an improved contrast. Size and shape were investigated with a stereo light microscope (Stemi 2000-C, Carl Zeiss, Oberkochen, Germany, ocular: W-PI, 10×/23, magnification: 5.0, 50 scale = 1 mm). Photographs were shot by mobile.

### 2.10. Sphericity

The sphericity of the pellet lots was measured by digital image processing (Camsizer^®^, Retsch, Haan, measuring particle size range 40–3000 µm, measured particles 20,000 per second). The chord length was used for the evaluation of particle size and particle size distribution.

### 2.11. SFV Measurement

The SFV probe was installed directly into the process chamber of the fluidized bed apparatus between the inner chamber wall and the Wurster inlet. Details are described elsewhere [27]. 

## 3. Results and Discussion

### 3.1. Properties of Sodium Benzoate and Polyacrylate Coated Pellets

SB pellets are received as a free-flowing material. The coating process runs without disturbances, and the special fluidized bed pattern with a Wurster inlet was homogeneous. The product shows a narrow particle size distribution [27]. The median x_50.3_ increases from 170 µm of uncoated Cellets^®^175 to 200 µm, and the sphericity of both initial Cellets^®^175 and SB pellets is above 0.9.

The polyacrylate coating of SB pellets is performed without undesired agglomeration; only very few twins and triplets are detected by microscopic observation (Figure 1). The median of polyacrylate pellets grows to 232.2 µm and the layer thickness to 16.1 µm (Table 4, P3, 17.6% polyacrylate content). The product yield losses and the incomplete sodium benzoate recovery derive from a material precipitation at the textile filter and the inner chamber wall. A sphericity above 0.9 indicates the existence of spherical products and a homogeneous processing.

### 3.2. Sodium Benzoate Release Kinetics

#### 3.2.1. Double Linear Diagram (Zero Order Release Kinetics)

After five minutes, more than 90% of the sodium benzoate is dissolved from SB pellets without a polymer layer due to a high solubility and a high dissolution rate. Showing properties of a strong electrolyte (sodium salt of benzoic acid with pK_a_ of 4.19, indicating a strong acid [31]), a considerable dissociation into sodium cations and benzoate anions takes place. The release from polyacrylate coated pellets is characterized by exponential curves (Figure 2). In general, the release rate decreases with the increasing polyacrylate layer thickness. The insoluble polyacrylate acts as a release barrier. After ten minutes, 30% sodium benzoate at low coating (P1), 20% at medium coating (P2) and 8% at high coating (P3) is released. The sodium benzoate dissolution rate is rather implausible as a release controlling process step. The diffusion of sodium benzoate molecules and sodium and benzoate ions out of the polymer layer forced by a high concentration gradient in the initial phase seems to be the rate controlling process. The release process starts with a high rate, and, in the terminal phase, the rate diminishes due to a nearly complete sodium benzoate release and a low concentration gradient across the polyacrylate film. In the case of the low polyacrylate coating, the CoD of the zero order kinetics amounts to 0.57 (Table 5, P1), giving no probability for zero order release kinetics at all. The diffusion process referring to first order kinetics seems to be the rate controlling step. Otherwise, the zero order CoD increases with the increasing polyacrylate layer thickness (P2: 0.70, P3: 0.93), indicating the growing influence and interaction of other processes like polymer swelling and retarded diffusion over a prolonged diffusion distance. With an increasing polyacrylate amount the release rate decreases, as indicated by the decreasing AUC, decreasing DE and increasing MDT (Table 6).

The release profiles of lots P1 and P2 (Figure 2) differ only slightly, so the f_1_ of 12 (Table 7) is in the range below 15 and indicates equivalence between P1 and P2; whereas the deviation of the profiles of P1/P3 and P2/P3 is much more pronounced due to thicker polyacrylate coating layers leading to an f_1_ above 15 and to the evaluation “not equivalent” regarding the relative error between both release profiles calculated from the cumulative released amounts R_i_ and T_i_ at certain moments. The increasing coating layer thickness leads to clearly different release profiles. 

The similarity factor f_2_ decreases with the increasing layer thickness and diminished release rate, which is obvious comparing P1 with P2 (74) and P1 with P3 (63). Nevertheless, both f_2_s confirm the similarity of the release profiles. 

Release curves (P1cal, P2cal and P3cal, Figure 2) were calculated according to first order kinetics (Equation (2)) and by use of the experimental release rate constants of P1, P2 and P3 (Table 5) from the Sigma minus plots (Figure 3).

The double linear drawn calculated curves (grey colour) meet roughly the corresponding experimental curves (black colour, Figure 2). The deviation of the experimental from the calculated curve is highest for P3 with the thick polyacrylate layer due to an increased coincidence of the following processes and circumstances: slow polyacrylate film wetting and swelling, slow water molecule uptake and diffusion through the polyacrylate film to the sodium benzoate layer, dissolution of sodium benzoate and diffusion through the swollen polymer film into the release fluid. The high thickness of the polyacrylate layer and therefore the long diffusion path and change of the sodium benzoate concentration gradient via the polyacrylate layer with the ongoing process is of important influence on the release rate. 

#### 3.2.2. First Order Kinetics, Sigma Minus Function

The Sigma minus function gives linear trends of the sodium benzoate release (Figure 3) according to first order kinetics and a CoD above 0.9 (Table 5). The release rate constant k_1_ decreases with growing polyacrylate coating (Table 8). The calculated curves P1cal and P2cal (Equation (3)) meet the experimental curves of P1 and P2, respectively (Figure 3). The diffusion of sodium benzoate through the polyacrylate layer and to some extent the polymer swelling are the rate controlling steps. The more pronounced deviation of the experimental from the calculated curve in case of P3 is explained by the reasons mentioned above.

#### 3.2.3. First Order Kinetics, Weibull Function

The Weibull function gives linear curves (Figure 4) comparable to the Sigma minus function (Figure 3) with coefficients of determination of 0.99 for P2 and P3 (higher polyacrylate coating) and a value of 0.87 for P1 (Table 5) due to the fast release in the initial phase and finally the slow release rate after 45 min (x-axis value 3.8, Figure 5).

A shape parameter of 1 indicates a monophasic and b > 1 a multiphasic release process with an initial lag time due to a wetting and a swelling of the polyacrylate film in the present case and an accelerated release rate up to the inflection point by high concentration gradient followed by a slower release rate due to a concentration gradient decrease when the drug dissolution and the release are finished. P1 with low coating gives nearly monophasic release kinetics (shape parameter 1.08, Table 8) whereas P2 (1.58) and P3 (1.36) give hints of a pronounced multiphasic release. The scale parameter (1/a) refers numerically to the rate constant and decreases with the increasing coating layer thickness (Table 8). The time parameter t_63_._2%_ is the moment when 63.2% of sodium benzoate is released. The value ascends with the increasing polyacrylate layer thickness from 30 to 70 min (Table 8 and Figure 5, compare also Figure 2).

The polyacrylate coating layer thickness (Table 4) proves to have a strong influence on the release kinetics. The manufacturing of coated pellet products with the intended drug release may be realized in the following way: pellet lots are manufactured with an increasing coating layer thickness in a preliminary step on a laboratory scale. The layer thickness is measured by the SFV probe. The in vitro drug release of these lots is investigated and correlated with the polymer coating layer thickness. The coating process in the production scale is interrupted when the desired coating layer thickness is detected.

#### 3.2.4. Square Root Function

The cumulative release plot versus the square root of time gives straight lines in case of a drug release by diffusion from non-disintegrating matrices like matrix tablets and semisolid systems (ointments, creams). Lots P1 and P2 show nearly straight lines in the time interval 10 to 60 min (Figure 6). The initial phase up to 10 min and the terminal phase after 60 min do not meet the square root model. CoDs reach values between 0.81 (P1) and 0.94 (P3, Table 5). This model is not suitable to describe the release profile in the present case of drug pellets with an insoluble but swellable polymer coating.

#### 3.2.5. Cubic Root Function

The cubic root function is valid for a dissolution of spherical particles due to reduction of weight and surface area. The difference of cubic roots of dose and cumulative released substance plotted versus time should give straight lines. This is not the case with the sodium benzoate release from polyacrylate coated pellets (Figure 7). The deviation from the linearity is pronounced for P1 and P2 in the terminal release phase after 45 min (CoD 0.68 and 0.80, respectively, Table 5) whereas the slow releasing P3 gives a curve with a DoD of 0.98. Only at a thick polyacrylate layer does the cubic root model seem to be suitable to describe the release kinetics, whereas the lower coated lots P1 and P2 do not refer to cubic root kinetics and dissolution of spheres.

## 4. Conclusions

Inert Cellets^®^175 were coated in a first step with the model drug sodium benzoate and in a second step with a water insoluble polyacrylate dispersion in a fluidized bed with a Wurster inlet. Particle size increase and coating layer thickness were measured in-line over the whole processes by the SFV probe and detected at each moment of the process. The in vitro sodium benzoate release was investigated and release profiles were linearized and evaluated with different kinetic models.

With the increasing polyacrylate coating layer thickness, the sodium benzoate release rate decreases, as evaluated by release parameters, release rate constants, AUC, MDT and DE. A difference factor f_1_ above 15 showed dissimilar release rate profiles for lower coated (P1, P2) compared with high coated (P3) polyacrylate SB pellets, indicating a significant influence of the coating layer thickness on the sodium benzoate release. The similarity factors f_2_ of 67 to 74 refer to similar release profiles of the lots P1, P2 and P3.

The high CoD of linearized sodium benzoate release profiles in the case of the polyacrylate coating predetermined first order kinetics as the most applicable, which is explained by the significant influence of the sodium benzoate diffusion through the swollen polyacrylate film outside of the pellets. With the increasing coating layer thickness, the polymer swelling, the long-distance diffusion process for both water and sodium benzoate and the increase of the concentration gradient become stronger influences on the release profiles.

The detailed investigation of the release rate profiles as dependent on the polymer coating layer thickness permits the detection of the coating process endpoint and the manufacturing of a custom-made coated drug pellet product with a defined drug release.

## Figures and Tables

**Figure 1 pharmaceutics-16-01307-f001:**
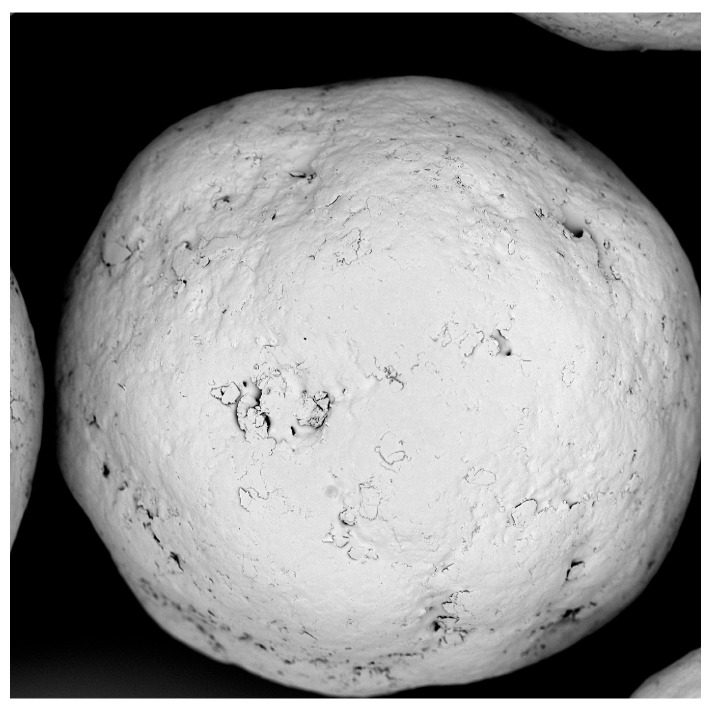
SEM photograph of a polyacrylate coated SB pellet.

**Figure 2 pharmaceutics-16-01307-f002:**
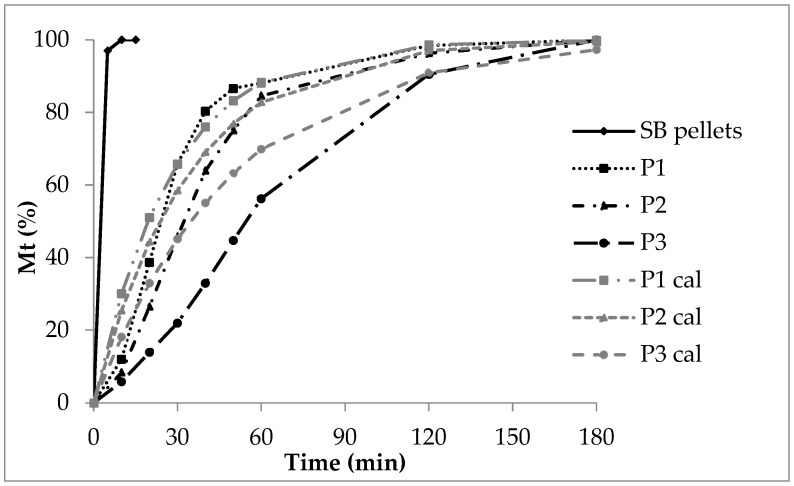
Double linear plot of the sodium benzoate release, SB pellets without polyacrylate layer, experimental release from polyacrylate-coated lots P1, P2 and P3 with increasing layer thickness and calculated release P1cal, P2cal and P3cal.

**Figure 3 pharmaceutics-16-01307-f003:**
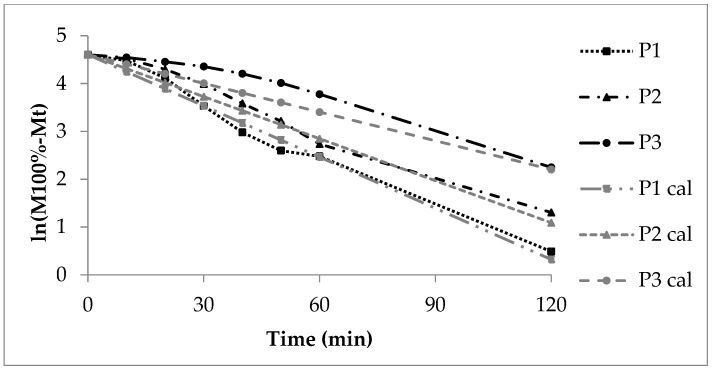
First order Sigma minus function of the experimental and calculated (cal) sodium benzoate release, lots P1, P2 and P3.

**Figure 4 pharmaceutics-16-01307-f004:**
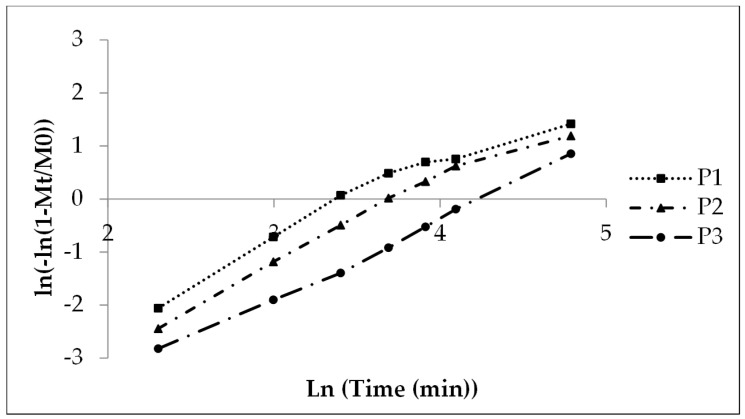
First order Weibull function of the experimental sodium benzoate release, lots P1, P2 and P3.

**Figure 5 pharmaceutics-16-01307-f005:**
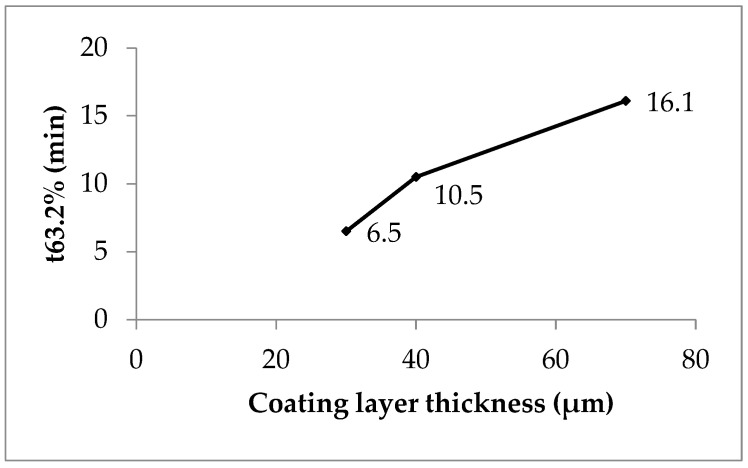
Weibull function release parameter t_63.2%_ versus coating layer thickness, lots P1 (coefficient of determination 0.87, polyacrylate content 6.5%), P2 (0.99, 10.5%) and P3 (0.99, 16.1%).

**Figure 6 pharmaceutics-16-01307-f006:**
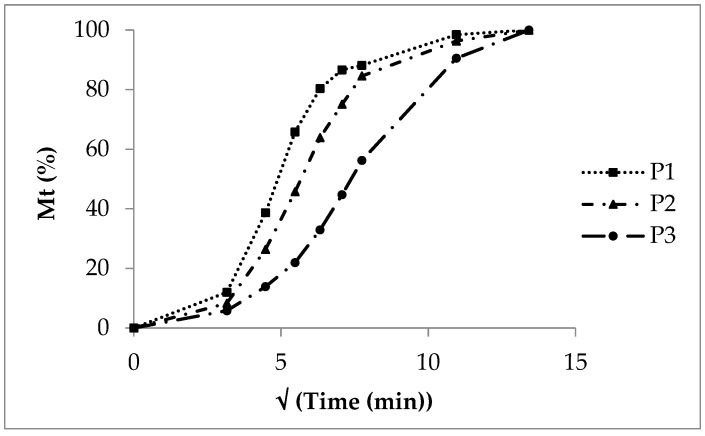
Square root function of the experimental sodium benzoate release, lots P1, P2 and P3.

**Figure 7 pharmaceutics-16-01307-f007:**
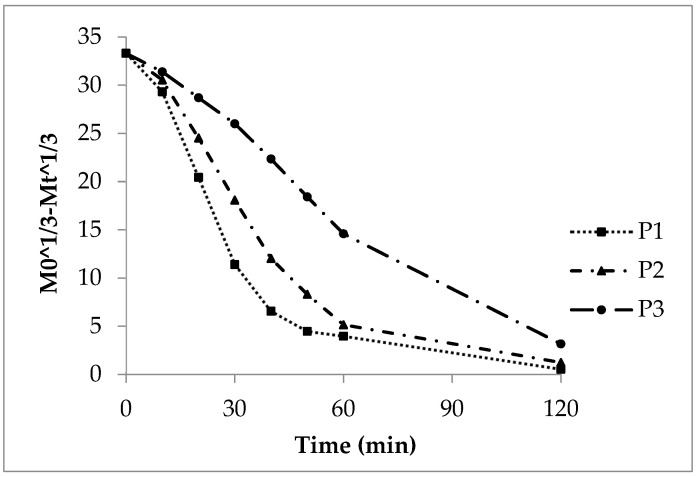
Cubic root function of the experimental sodium benzoate release, lots P1, P2 and P3.

**Table 1 pharmaceutics-16-01307-t001:** Sodium benzoate pellet formulation.

	Content (%)
Sodium benzoate	32.6
Microcrystalline cellulose	65.3
PVP	1.6
Talcum	0.5
	100.0

**Table 2 pharmaceutics-16-01307-t002:** Polyacrylate coated sodium benzoate pellet.

Lot	P1	P2	P3
	Content (%)		
Sodium benzoate	25.9	24.4	22.9
Microcrystalline cellulose	55.7	52.6	49.3
PVP	1.3	1.2	1.1
Polyacrylate	11.1	14.3	17.6
Talcum	4.9	6.1	7.4
Magnesium stearate	1.1	1.4	1.7
	100.0	100.0	100.0

**Table 3 pharmaceutics-16-01307-t003:** Process parameters of pellet fluidized bed coating with sodium benzoate (first step) and polyacrylate (second step).

Parameter	First Step	Second Step
	Sodium benzoate	Polyacrylate
Pellet batch (g)	300	
Process air temperature (°C)	80	40
Product temperature (°C)	40	25
Process air volume rate (m^3^/h)	40–60	
Spray rate (g/min)	20	6
Spray pressure (bar)	3	

**Table 4 pharmaceutics-16-01307-t004:** Median, polyacylate layer thickness, product yield, sodium benzoate content and sphericity of polyacrylate coated SB pellets.

	X_50.3_ (µm)	Polyacrylate Layer Thickness (µm)	Yield (%)	Sodium Benzoate Content (%)	Sphericity (-)
P1	213.0	6.5	84	92	0.91
P2	221.0	10.5			
P3	232.2	16.1			

**Table 5 pharmaceutics-16-01307-t005:** CoD of sodium benzoate release profiles, kinetic models of zero order, first order, square root and cubic root; lots P1, P2 and P3.

	CoD (R^2^)		
Model	P1	P2	P3
Zero order	0.57	0.70	0.93
First order Sigma minus	0.98	0.98	0.95
First order Weibull	0.87	0.99	0.99
Square root	0.81	0.88	0.94
Cubic root	0.68	0.80	0.98

**Table 6 pharmaceutics-16-01307-t006:** Area under the curve, AUC, dissolution efficiency, DE, and mean dissolution time, MDT, of sodium benzoate release; lots P1, P2 and P3.

	AUC (%∗min)	DE (-)	MDT (min)
P1	14,820	0.82	32
P2	13,927	0.77	41
P3	11,587	0.64	63

**Table 7 pharmaceutics-16-01307-t007:** Difference factor and similarity factor of sodium benzoate release profiles, comparison of lots P1, P2 and P3.

Parameter	Evaluation	P1/P2	P1/P3	P2/P3
Difference factor f1	“equivalent” 0–15	12	24	25
Similarity factor f2	“similar” 50–100	74	63	67

**Table 8 pharmaceutics-16-01307-t008:** First order release parameters of the Sigma minus and Weibull function, lots P1, P2 and P3.

	k_1_ (1/min) Sigma Minus	b (-) Weibull	1/a (-) Weibull	t_63.2%_ (min) Weibull
P1	0.036	1.08	0.25	30
P2	0.030	1.58	0.17	40
P3	0.020	1.36	0.17	70

## Data Availability

All relevant data are available in the article.

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
