# Peer review of "Influence of Polymer Film Thickness on Drug Release from Fluidized Bed Coated Pellets and Intended Process and Product Control"

_pharmaceutics, 2024, doi:10.3390/pharmaceutics16101307_

Round 1
Reviewer 1 Report
Comments and Suggestions for Authors
I recommend this manuscript for publication after minor, mainly technical, revisions are made.
-please specify novelty of your work in introductory part. What did you do different and/or better in comparison to the other groups
-term “coating degree” is vague and weird, hast to be eliminated
-Fig.1 is too generic. Perhaps, add coating cartoon or, otherwise, just eliminate. In the way it is shown, no additional information is gathered
-Fig.2 has to be eliminated. This photo does not have much merit.
-please add standard deviations throughout the manuscript.
-title is very generic, especially “influence of polymer films…” saying. Please rephrase to be more specific (may be thickness??)
Comments on the Quality of English Languagenone
Author Response
Thank you very much for taking the time to review this manuscript. Please find the respond to the comments and the revised manuscript in attachments.

Reviewer 2 Report
Comments and Suggestions for Authors
Figure 1: "The scheme of the two-layer pellet should be made more attractive with the help of a 3D drawing tool (such as Google SketchUp).
Although, the analysis of in vitro dissolution profiles of encapsulated drugs usually involves the linearization of the applied mathematical model, in this way the transformation of the measured data. This transformation was necessary when the data evaluation was done manually, since the spread of personal computers, it is absolutely unnecessary. There is a freely available and free of charge a spreadsheet routine could provide a universal and reproducible data analysis strategy without the need for linearization. You can find an download from here:
https://www.sciencedirect.com/science/article/pii/S0167732221001318
Download/ Supplementary material 2/Download spreadsheet (2MB)
Author Response

(The authors gave the same response as above.)
